# The Role of Novel Motorized Spiral Enteroscopy in the Diagnosis of Cecal Tumors

**DOI:** 10.3390/diseases10040079

**Published:** 2022-10-04

**Authors:** Amir Selimagic, Ada Dozic, Azra Husic-Selimovic

**Affiliations:** 1Department of Gastroenterohepatology, General Hospital “Prim. dr. Abdulah Nakas”, 71 000 Sarajevo, Bosnia and Herzegovina; 2Department of Internal Medicine, General Hospital “Prim. dr. Abdulah Nakas”, 71 000 Sarajevo, Bosnia and Herzegovina

**Keywords:** motorized spiral enteroscopy, small bowel disease, gastrointestinal tumor

## Abstract

Small bowel and ileocecal diseases remain a diagnostic and therapeutic challenge, despite the introduction of various modalities for deep enteroscopy. Novel Motorized Spiral Enteroscopy is an innovative technology that uses an overtube with a raised spiral at the distal end to pleat the small intestine. It consumes less time and meets both the diagnostic and therapeutic needs of small bowel diseases. The objective of this article is to highlight the possibility of using NMSE as an alternative technique when a target lesion is inaccessible during conventional colonoscopy or cecal intubation cannot be achieved. We report the case of a 61-year-old man who presented with pain in the right lower abdominal segment, diarrhea, and rapid weight loss for more than 3 months. An initial ultrasound showed a suspicious liver metastasis. Computerized tomography scans showed an extensive ileocecal tumor mass with liver metastasis. The colonoscopy was unsuccessful and incomplete due to dolichocolon and intestinal tortuosity. Later, endoscopy was performed using a Novel Motorized Spiral Enteroscope in a retrograde approach, passing the scope through the anus and colon up to the ileocecal segment, where a tumor biopsy was performed and adenocarcinoma was pathohistologically confirmed.

## 1. Introduction

The role of endoscopy in the diagnosis and treatment of gastrointestinal disorders has become irreplaceable in the past decades. Even though new endoscopic modalities have been developed, distant parts of the gastrointestinal system, especially the small bowel, remain inaccessible by endoscopic methods in some cases. In 2001, video capsule endoscopy was presented as an innovative technology that enabled examination of the entire small bowel but with a significant drawback regarding the impossibility of performing therapeutic interventions [1]. Further development of deep enteroscopy was made with the introduction of double-balloon (DBE) endoscopy, and later on, single balloon endoscopy (SBE) with the push and pull technique [1,2]. However, these techniques also have certain limitations, such as long procedure duration, and, in some cases, inefficiency in achieving a complete examination of the small bowel [3,4,5]. Spiral enteroscopy is a technique that uses a spiral tube at the bottom end of the endoscope allowing diagnostic and therapeutic access to obscure parts of the digestive tract. It was used for the first time in 2008. However, it has been found to be a time-consuming and complex procedure since it requires two operators and a manual rotation [6,7,8].

Novel Motorized Spiral Enteroscopy (NMSE) is a new technology with an incorporated motor adjacent to the endoscope. The spiral enteroscope control unit regulates the direction and speed of rotation of the spiral segment (Figure 1) [8,9]. In addition, it is performed by a single operator.

In 2016, Neuhaus et al. [10] reported the first clinical use of this procedure for the treatment of angiodysplasias in the jejunum of a 48-year-old patient with iron-deficiency anemia. Since then, NMSE has become the focus of research as a potential solution for the limitations of currently available device-assisted enteroscopy techniques. This technique is considered effective and safe and allows for performing both diagnostic and therapeutic interventions on the small bowel. It also has a shorter duration compared to other deep enteroscopy modalities [11]. The most common indications for NMSE include obscure gastrointestinal bleeding and suspected inflammatory bowel disease. In addition, it is used for therapeutic interventions such as polypectomy, hemostasis, or stricture dilatation [6].

Since this is a newly introduced diagnostic procedure that is used only in a small number of facilities, the available literature on this topic is very limited. In addition to the small intestine, the use of this device via the retrograde route can be of great importance in the diagnosis of ileocecal diseases when the target lesions are difficult to access with conventional colonoscopy. According to the literature, 1.6–16.7% of all colonoscopies are found to be unsuccessful due to failure to achieve cecal intubation [12,13,14]. The most frequent causes of incomplete colonoscopy are stenosis or inadequate bowel cleansing, previous abdominal or pelvic surgery, female sex, older age, low body mass index, and diverticulosis [12,15,16]. In addition, an abnormally long, tortuous colon is considered a significant factor that may result in a failed cecal intubation [17]. In 4.3% of patients who underwent incomplete colonoscopy, an advanced tumor was missed [12,18]. This finding highlights the significance of the visualization of the entire colon during the examination. Therefore, NMSE could be a promising and effective alternative technique to achieve the visualization of the entire colon and access to target lesions. Here, we report the case of a patient with a tumor of the terminal ileum and caecum who was not a candidate for biopsy via colonoscopy due to dolichocolon.

## 2. Case Presentation

A 61-year-old man presented to the hospital with a history of pain in the right groin area, a change in his bowel habits with frequent mushy stools, and unintentional weight loss. These symptoms started about 3 months prior to admission. He was a professional driver with a history of type 2 diabetes mellitus and arterial hypertension who did not consume alcohol or tobacco. No family history of cancer or other inherited medical conditions were reported. Physical examination revealed mild pain to palpations in the right upper and lower quadrants of the abdomen. There were no other abnormal findings during the examination.

The initial evaluation included laboratory tests, which revealed elevated tumor markers CEA 3410 μg/L (upper reference limit 3.4 μg/L) and CA 19-9 292 U/mL (upper reference limit 25 U/mL), whereas AFP was normal (1.23 kU/L); upper reference limit 5.8 kU/L). The other laboratory findings were normal.

The initial abdominal ultrasound verified changes in the liver parenchyma suspected of metastatic lesions. The abdomen and pelvis CT scans showed multiple focal nodal areas of the liver parenchyma highly suspicious for liver metastases. In the area of the caecum, there was an infiltrative thickening of the wall of the terminal ileum area with associated luminal narrowing, reactive changes of perivisceral adipose tissue, and reactive lymphadenopathy (Figure 2). The proximal endoscopy findings were normal.

Colonoscopy was incomplete since it was performed only up to hepatic flexure. Further progression to CT-verified cecal infiltration was unsuccessful due to an abnormally long, large intestine (dolichocolon) and malignant rotation of the intestine. NMSE was performed in a retrograde approach in order to reach the ileocecal infiltration. A spiral enteroscope was used to examine the colon because of the abnormal tortuosity of the intestine due to a tumor of the ileocecal segment as well as dolichocolon. A series of light movements of the spiral enteroscope enabled passing through the lumen of the large bowel and reaching a region with circular infiltration that almost completely obstructed the lumen (Figure 3).

Six biopsies were taken and adenocarcinoma was pathohistologically confirmed (Figure 4). The patient was presented to a multidisciplinary team to decide on further treatment.

## 3. Discussion

The small bowel and ileocecal segment diseases accompanied by pathological intestinal tortuosity have always been a significant drawback for endoscopists. In the last two decades, with the introduction of DBE in 2001 and SBE in 2006, enteroscopic examination has significantly improved [1,19,20,21]. Video capsule endoscopy was found to be useful for the visualization of the entire small bowel. However, it does not allow the performing of therapeutic interventions and biopsies, which represents a significant limitation of this procedure [22]. The NMSE is a revolutionary endoscopic tool for the diagnosis and therapy of distant areas of the gastrointestinal tract that are inaccessible by proximal and distal endoscopy. Therefore, it is extremely useful for examination and interventions in the deep parts of the small bowel [23]. In this case, NMSE enabled a complete visualization of the colon and reached the ileocecal infiltration quickly and efficiently by “pleating” the abnormally long, tortuous colon. In addition, this technique allowed the performing of control movements of the endoscope and provided a more stable position for obtaining representative biopsies compared to the conventional colonoscope.

The available literature mainly deals with the usage of this procedure for the diagnosis and treatment of small bowel diseases since it has been primarily developed for that purpose. However, some authors have reported the benefits of using this technique in cases when conventional colonoscopy is unsuccessful.

In a recent study, the authors reported about 36 patients who had previously undergone incomplete diagnostic and/or therapeutic colonoscopies due to dolichocolon [12]. The cecal intubation rate was 100% and the median cecal intubation time was 10 min (range 4–30). The diagnostic yield was 64%, and neoplastic lesions were found in 23 patients. No adverse events were noted. They concluded that NMSE is an effective alternative for diagnostic and therapeutic colonoscopy in patients with dolichocolon. To the best of our knowledge, our case represents the first use of this device after an incomplete colonoscopy due to dolichocolon in our country and region.

Beyna et al. [24] evaluated the efficacy and safety of NMSE for diagnostic colonoscopy in 30 patients. Diverticulosis was the most common finding (43.3%) and the average procedure time was 20.8 min (range 11.4–55.3). No severe adverse events occurred. According to these results, NMSE is also considered safe and feasible for diagnostic colonoscopy.

In other studies, this procedure was mainly performed using anterograde and/or retrograde approaches in patients with gastrointestinal bleeding and unrevealing proximal and distal endoscopy and in cases when small bowel disease was suspected [25,26]. Prasad et al. [6] reported a case series of 14 patients in whom NMSE was performed in anterograde, retrograde, or both approaches, and the most common findings were strictures or ulcers of the jejunum and ileum. Ramchandani et al. [27] reported about 61 patients who underwent NMSE due to symptomatic small bowel disease. Total enteroscopy was performed in 60.6% of patients, for whom 29.5%, both anterograde and retrograde approaches were used. Singh et al. [28] evaluated 54 patients for small-bowel disease using NMSE. They have also used this technique in both anterograde and retrograde approaches. The retrograde approach was performed when the anterograde approach was contraindicated and in patients with previous imaging findings that showed suspected lesions within 150 cm of the ileocecal valve [28]. The average duration of a procedure performed in a retrograde approach was 35 min in a study by Ramchandi et al. and 90 min in a study by Prasad et al. [6,27].

Since NMSE is a new technology, data are lacking on its efficacy and safety in comparison with other endoscopic modalities. According to the literature, cecal intubation was achieved in 88–95% using DBE [29,30,31], 100% using short DBE [32], 93–100% for SBE [33,34,35], and 92% for manually driven spiral overtube-assisted colonoscopy [36]. The diagnostic yield of DBE was 68.1% [1,37] and for SBE, it was 47 to 60% [38,39,40]. The diagnostic and therapeutic yields of the NMSE were 80% and 86.7%, respectively [23].

In addition, recent research has shown that spiral enteroscopy could be an effective solution for ERCP in patients with altered gastrointestinal anatomy (after gastric and duodenal resection) and it could significantly shorten the time required to perform the procedure [41].

So far, rare adverse events associated with NMSE have been reported, including acute pancreatitis, perforations, and erosive lesions of the esophageal mucosa [24]. No adverse events were encountered in our case as well as in other recent studies and case reports [12,24,27]. However, Prasad et al. reported major adverse events, which included hypothermia (3 out of 14 patients) and pancreatitis (1 out of 14 patients) [6].

In this report, we aimed to highlight the possibility of performing NMSE in patients after unsuccessful conventional diagnostic or therapeutic colonoscopy, especially in patients with dolichocolon and suspected ileocecal disease. As already understood, failure to achieve cecal intubation during the examination of the colon increases the risk of missing a lesion, especially neoplasms in the inaccessible parts of the bowel. Therefore, the visualization of the entire colon is necessary and when it cannot be achieved during conventional colonoscopy, NMSE should be performed as a safe and effective alternative technique. It could be considered an alternative technique in patients who have undergone incomplete colonoscopy due to different causes, mainly in patients with a long, tortuous colon, presence of angulation or fixation of bowel loops, as well as adhesions due to previous surgeries. In these cases, NMSE is effective because it enables the straightening of the loops by pleating the bowel. This mechanism allows for the deeper advancement of the endoscope through the colon and provides a more stable position for the device while performing polypectomies and biopsies compared to a conventional colonoscope. This technique should not be applied in cases where strictures of the bowel are present and cause severe narrowing of the intestinal lumen. In these cases, if increased resistance is detected, spiral rotation is stopped automatically to avoid perforation. In addition, the larger caliber of the spiral overtube and the rigidity of the attachment segment can increase the risk of perforation.

Additional studies with a larger number of patients are needed to examine the safety and efficacy of NMSE and to compare this new technology with conventional colonoscopy and balloon-associated enteroscopy.

## 4. Conclusions

The NMSE is a procedure that enables both diagnostic and therapeutic interventions for small bowel diseases. However, this case report highlights the possibility of using this innovative tool as an alternative technique when the target lesion is difficult to access during conventional colonoscopy. Its advantages include a simplified technique performed by a single operator, a shorter time of examination, and the ability to perform both diagnostic and therapeutic interventions.

## Figures and Tables

**Figure 1 diseases-10-00079-f001:**
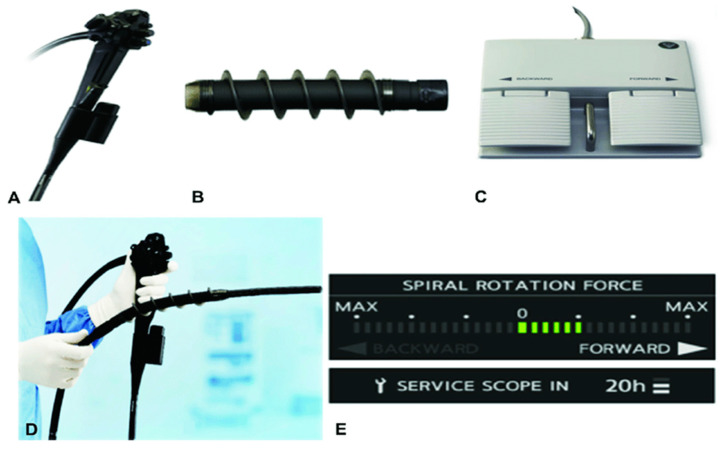
(**A**) Integrated motor in motorized spiral enteroscope. (**B**) Power spiral tube. (**C**) Foot switch with forward and backward pedals. (**D**) Power spiral enteroscope. (**E**) Force gauge on power spiral control unit for visual indication of the rotational direction. Pictures courtesy of Olympus Corp, Tokyo, Japan.

**Figure 2 diseases-10-00079-f002:**
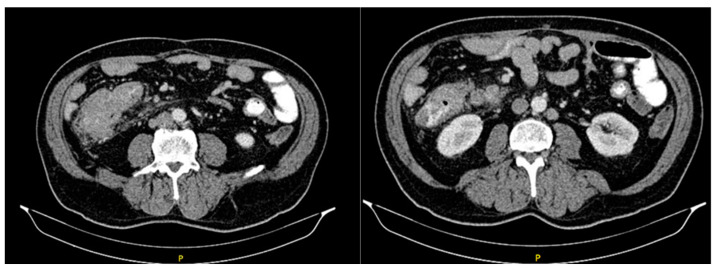
Abdominal CT scan showing tumor mass.

**Figure 3 diseases-10-00079-f003:**
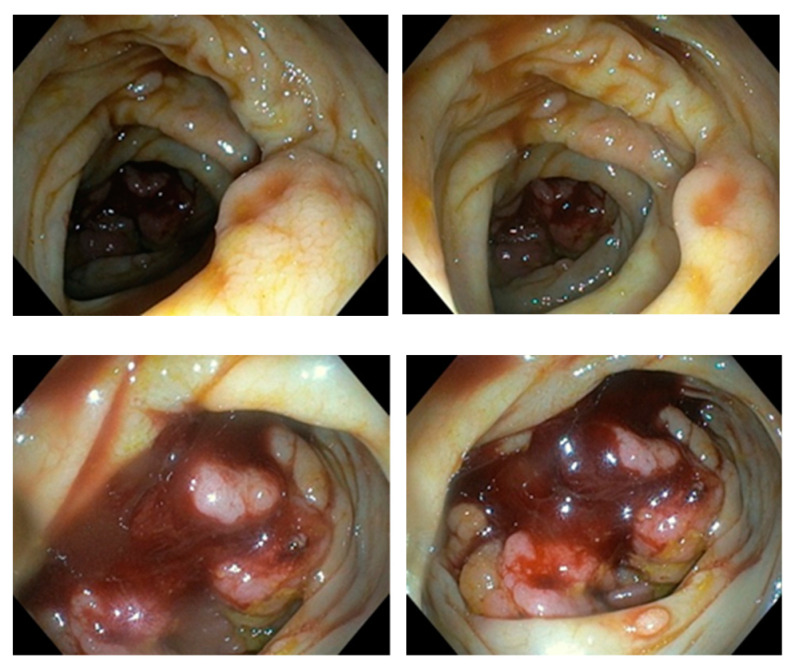
Endoscopic picture of cecal region tumor by Novel Motorised Spiral Enteroscope.

**Figure 4 diseases-10-00079-f004:**
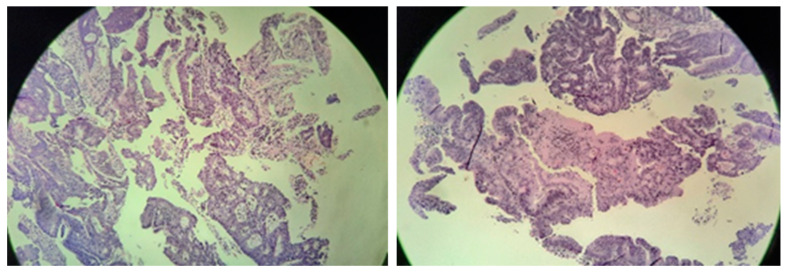
Microscopic images of cecal adenocarcinoma.

## Data Availability

Not applicable.

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
