# Peer review of "The Role of Novel Motorized Spiral Enteroscopy in the Diagnosis of Cecal Tumors"

_diseases, 2022, doi:10.3390/diseases10040079_

Round 1

Reviewer 1 Report

This case report describes the potential role of Novel Motorized Spiral Enteroscopy in the diagnosis of cecal tumors. The manuscript is well written, easy to read and understand. Since this diagnostic modality is still under-researched, the paper deserves to be published in the "Diseases" journal. However, this reviewer has a few minor suggestions:

1.      Page 1, line 5 – “It consumes less time and suits both the diagnostic and therapeutic need …” should be changed to “It consumes less time and meets both the diagnostic and therapeutic needs…”

2.      Page 1, line 27 – “digestive diseases” should be changed to “gastrointestinal disorders”

3.      Page 1, line 40 – “However, it is found to be a time-consuming and complex procedure since it requires 2 operators and a manual rotation [6-8]” should be changed to “However, it has been found to be a time-consuming and complex procedure since it requires two operators and manual rotation [6-8].”

4.      Page 1, line 44 – “It is performed by a single operator.” – In order for the sentence not to sound stiff, add a conjunction at the beginning, e.g. “Also”

5.      Page 2, line 51 – “…and safe, and it allows…” should be changed to “…and safe and allows…”

6.      Page 2, lines 51-57 – This paragraph should be rearranged more chronologically. I would start this paragraph in the following way: “In 2016, Neuhaus et al. [11] reported the first clinical use of this procedure for the treatment of angiodysplasias in the jejunum of 48-year-old patient with iron-deficiency anemia. Since then, NMSE has become the focus of research as a potential solution for the limitations of currently available device-assisted enteroscopy techniques. This technique is considered effective and safe and allows performing both diagnostic and therapeutic interventions of small bowel. It also has a shorter duration compared to other deep enteroscopy modalities [10]. The most common indications for NMSE include obscure gastrointestinal bleeding and suspected inflammatory bowel disease. Besides, it is used for therapeutic interventions such as polypectomy, hemostasis, or stricture dilatation [6].”

7.      Page 2, lines 59-60 – “Furthermore, it is mainly used for diagnostic and therapeutic interventions in the small bowel” - This sentence is completely the same as the one found on page 2 of line 51-52 (“This technique is considered effective and safe, and it allows performing both diag- 51 nostic and therapeutic interventions of small bowel.”) However, if you want to combine this sentence with the next one in order to emphasize the additional diagnostic possibilities of this technique, it should be changed to: “In addition to the small intestine, the use of this device via the retrograde route can be of great importance in the diagnosis of ileocecal diseases when the target lesions are difficult to access with conventional colonoscopy”

8.      Page 2, lines 62-63 – “…due to a failure to achieve cecal intubation…” – please, delete “a”

9.      Page 2 and 3, lines 77-80 – “He stated…”, “He reported…”, “He denied…”, “He is…” – Please rephrase these sentences (you can combine two or even all together into one) in order to avoid repeating the pronoun "He" over and over again.

10.  Page 3, line 84 – Please, add units after the tumor marker values.

11.  These two sentences - “Since this is a newly introduced diagnostic procedure, used only in a small number of facilities, the available literature on this topic is very limited.” and “Since this is a relatively new diagnostic and therapeutic tool, there is a lack of literature on this topic.” are the same. Please, delete one of them or combine a sentence from the "Discussion" (“Since this is a relatively new diagnostic and therapeutic tool, there is a lack of literature on this topic.”) with the sentence that comes next “The available literature mainly deals with the usage of this procedure 127 for the diagnosis and treatment of small bowel diseases.”

12.  Page 5, line 141 – “In the other studies,…”– please, delete “the”

13.  Page 5, line 144 – “…in which NMSE was performed in anterograde, retrograde, or both approaches…“ should be changed to“…in whom NMSE was performed using an anterograde, retrograde, or both approaches…”

14. Page 5, line 166 – “In our case, no adverse events were encountered. No serious adverse events were reported in other recent studies and case reports” please, merge this two sentences in one.

Reviewer 2 Report

Comments to the Author

This is an interesting, however below are some questions and comments.

Comments

1.  Please add a comment on the reason why complete colonoscopy was successful with NMSE in this case.

2. The usefulness of NMSE for failed diagnostic and/or therapeutic colonoscopy has been already reported.  Please try to explain the new findings in this report.

3. Please also add a description as to in what kind of failed colonoscopy cases NMSE should be applied and useful.

Author Response

Response to Reviewer 2 Comment

Comment: This is an interesting, however below are some questions and comments.

Response: Thank you for your time and effort that you have dedicated in reviewing this manuscript. We really appreciate all of your comments and suggestions.

Comment 1: Please add a comment on the reason why complete colonoscopy was successful with NMSE in this case.

Response 1: Thank you for your suggestion. In this case, NMSE enabled complete visualization of the colon and reached the ileocecal infiltration by “pleating” the abnormally long, tortuous colon. In addition, this technique allowed performing control movements of the endoscope and provided a more stable position for obtaining representative biopsies compared to the conventional colonoscope. We have added this explanation in the manuscript (Page 4, lines 127-131).

Comment 2: The usefulness of NMSE for failed diagnostic and/or therapeutic colonoscopy has been already reported.  Please try to explain the new findings in this report.

Response 2: Thank you for your comment. There is a lack of literature on the usefulness of NMSE in terms of previously unsuccessful conventional colonoscopy due to dolichocolon, especially in cases with associated ileocecal infiltration. We have found only one study, published in August 2022, in which this technique has been performed in patients who had previously undergone incomplete colonoscopy due to abnormally long colons. Furthermore, to the best of our knowledge, this case represents the first use of this device after incomplete colonoscopy due to dolichocolon in our country and region.

In this report, we aimed to highlight the possibility of performing NMSE after unsuccessful conventional diagnostic or therapeutic colonoscopy. As it is known, failure to achieve cecal intubation during examination of the colon increases the risk of missing a lesion, especially neoplasms in the inaccessible parts of the bowel. Therefore, visualization of the entire colon is necessary and when it can not be achieved during conventional colonoscopy, NMSE should be performed as a safe and effective alternative technique. Hence, we find that this report makes a valuable contribution to the recognition of NMSE potential role in order to apply it more often in everyday practice. We have added this explanation in the manuscript (Page 5, lines 179-185).

Comment 3: Please also add a description as to in what kind of failed colonoscopy cases NMSE should be applied and useful.

Response 3: Thank you for your suggestion. The NMSE could be considered as an alternative technique in patients who have undergone incomplete colonoscopy due to different causes, mainly in patients with long, tortuous colon, presence of angulation or fixation of bowel loops, as well as adhesions due to previous surgeries. In these cases, NMSE is effective because it enables straightening the loops by pleating the bowel. This mechanism allows deeper advancement of the endoscope through the colon and provides a more stable position of the device while performing polypectomy and biopsy compared to a conventional colonoscope. This technique should not be applied in cases where strictures of the bowel are present and cause severe narrowing of the intestinal lumen. In these cases, if increased resistance is detected, spiral rotation is stopped automatically to avoid perforation. Also, the larger caliber of the spiral overtube and the rigidity of the attachment segment would increase the risk of perforation. We have added this description in the manuscript (Page 5, lines 185-196).